# FtsZ Interactions and Biomolecular Condensates as Potential Targets for New Antibiotics

**DOI:** 10.3390/antibiotics10030254

**Published:** 2021-03-04

**Authors:** Silvia Zorrilla, Begoña Monterroso, Miguel-Ángel Robles-Ramos, William Margolin, Germán Rivas

**Affiliations:** 1Centro de Investigaciones Biológicas Margarita Salas, Consejo Superior de Investigaciones Científicas (CSIC), 28040 Madrid, Spain; michel.robles@cib.csic.es (M.-Á.R.-R.); grivas@cib.csic.es (G.R.); 2Department of Microbiology and Molecular Genetics, McGovern Medical School, University of Texas, Houston, TX 77030, USA; William.Margolin@uth.tmc.edu

**Keywords:** bacterial cell division, FtsZ association states, protein interactions, biomolecular condensates, macromolecular crowding, phase separation, cytomimetic media, persister states

## Abstract

FtsZ is an essential and central protein for cell division in most bacteria. Because of its ability to organize into dynamic polymers at the cell membrane and recruit other protein partners to form a “divisome”, FtsZ is a leading target in the quest for new antibacterial compounds. Strategies to potentially arrest the essential and tightly regulated cell division process include perturbing FtsZ’s ability to interact with itself and other divisome proteins. Here, we discuss the available methodologies to screen for and characterize those interactions. In addition to assays that measure protein-ligand interactions in solution, we also discuss the use of minimal membrane systems and cell-like compartments to better approximate the native bacterial cell environment and hence provide a more accurate assessment of a candidate compound’s potential in vivo effect. We particularly focus on ways to measure and inhibit under-explored interactions between FtsZ and partner proteins. Finally, we discuss recent evidence that FtsZ forms biomolecular condensates in vitro, and the potential implications of these assemblies in bacterial resistance to antibiotic treatment.

## 1. Introduction

Resistance to antibiotics is already a major health threat and predicted to worsen unless new strategies are discovered to combat bacterial infections. Efforts in this direction include the identification of new potential antibiotic targets. The well characterized bacterial cell division machinery has lately attracted considerable attention in the quest for these new targets, as reflected by the growing number of reports on the subject [1]. This high level of interest is spurred by the general conservation of cell division proteins in most bacterial species, clear differences from proteins involved in cytokinesis of animal cells, and emerging detailed understanding of the regulatory and structural mechanisms underlying the cell division process in several model bacterial systems.

Bacterial cell division is orchestrated by the divisome, a dynamic multiprotein complex that coordinates partitioning of daughter chromosomes, localized cell wall synthesis, and membrane invagination in order to achieve robust and reliable separation of daughter cells [2]. Within the divisome, the FtsZ GTPase is considered the central protein of the cytokinesis machinery, as it forms a so-called Z-ring where the other components bind at the division site (Figure 1). Although FtsZ shows high structural similarity with the eukaryotic cytoskeletal protein tubulin, their amino acid sequences are less than 20% identical [3], and there are important differences in their GTP binding sites, polymerization features, and protein partners [4,5]. Among bacteria and archaea that carry it, FtsZ is largely conserved, with 40–50% sequence identity [6]; FtsZ orthologs are also present in plastids of algae and plants [1]. Therefore, this protein is a good target for antimicrobials, both for the low probability of cytotoxicity through tubulin cross-inhibition and for potential broad-spectrum activity.

FtsZ function relies on its ability to assemble into GTP-dependent polymers and to interact with other divisome proteins. The FtsZ monomer comprises a highly conserved N-terminal globular domain that contains the GTP binding site, an unstructured and variable linker, and a C-terminal domain that includes a short conserved tail [7]. Crucial interactions with other divisome proteins are mediated through the two conserved domains. In *Escherichia coli*, the conserved C-terminal tail binds to negative spatial regulators of FtsZ assembly such as MinC [8,9,10] and SlmA [11,12,13] (Figure 1a,f) as well as the two FtsZ membrane anchors FtsA [14,15,16] and ZipA [14,17,18,19] (Figure 1b) and also ClpX, one of the components of the protease complex ClpXP that degrades FtsZ [20,21] (Figure 1d). FtsZ also interacts with ZapA [20], a protein highly conserved across bacterial species that, together with ZapB and the DNA binding protein MatP, forms a complex involved in Z-ring positioning called the Ter-linkage [22] (Figure 1e).

Self-association of FtsZ leads to different types of higher-order structures depending on the conditions (Figure 2). Single-stranded polymers are formed through head-to-tail interactions between globular regions of two subunits [23]. Triggered by GTP binding, polymerization occurs above a critical concentration of protein [24], following a cooperative association mechanism [25]. Active GTPase sites are formed at the contact interface between two monomers [7]. The sizes of the filaments, which remain assembled until GDP accumulates due to GTP hydrolysis, depend on the conditions [26,27,28,29,30] and they laterally associate into bundles in crowded environments [31,32,33]. There is extensive literature available on the polymerization of FtsZ (see, for instance, [25]) and its inhibition, which leads to suppression of cell division and cell death [34].

When bound to GDP, FtsZ forms non-cooperative isodesmic oligomers whose size is modulated by salt and magnesium [35], and which increase in length in physiological crowding conditions [36]. In vivo, this GDP form of FtsZ is likely transient because of rapid exchange with GTP, except during starvation conditions. The GDP form of FtsZ is active in recognition of other proteins, including the aforementioned ZipA, MinC, and SlmA [37,38,39]. FtsZ-GDP is also a key player in the sequestration mechanisms used by FtsZ polymerization antagonists like the DNA damage-inducible SulA protein or the Kil peptide expressed by infecting bacteriophage λ [40,41] (Figure 1c,d).

In addition to assembling as polymers, it was recently shown that FtsZ-GDP can also form dynamic phase-separated condensates (defined as membraneless assemblies in which biomolecules concentrate) in vitro under conditions that mimic the crowded cell cytoplasm [42]. Formation of these condensates is facilitated by the interactions between FtsZ and its antagonist SlmA, which itself is bound to DNA. FtsZ can also form these condensates on its own (Figure 2), albeit less efficiently [43]. The assembly of biomolecular condensates arising from phase separation is emerging as a new mechanism for the spatial organization of biomolecules to optimize their function [44]. It provides a new framework to explain biological processes, as well as clues to decipher the mechanisms underlying major human diseases such as neurodegeneration, cancer, and infections, which may feature dysregulation of condensate formation [45]. Although biomolecular condensates were initially described only in eukaryotic cells, recent research shows bacterial proteins can also assemble into these dynamic membraneless compartments in vivo [46,47]. For example, proteins involved in essential bacterial processes such as chromosome segregation, DNA compaction and repair, mRNA transcription, or mRNA degradation have been shown to form condensates in vitro and in vivo [47].

How might antibacterial compounds antagonize FtsZ? One mechanism would simply be to hinder FtsZ self-association [1,48,49]. Indeed, most anti-FtsZ compounds based on rational drug design perturb FtsZ polymerization, by blocking key sites for GTP binding or FtsZ subunit contacts [48]. Another mechanism might be to hyperstabilize or promote inappropriate bundling of FtsZ polymers, which would perturb polymer dynamics and “freeze” them in nonfunctional states [48]. In contrast to targeting of FtsZ polymerization, the heterotypic interactions of FtsZ with other proteins and its indirect binding to the membrane have been relatively under-explored as drug targets. Inhibiting these protein-protein interactions may help to reduce the impact of resistance [1,50]. Furthermore, because FtsZ’s binding partners are not as well conserved, inhibition of these interactions would be more species specific [48]. The reported tendency of FtsZ complexes to undergo crowding-induced dynamic condensation may provide an additional opportunity for therapeutic intervention. Interestingly, condensate formation is often observed in cells growing under stress conditions and thus correlates with the emergence of persisters, a sub-population of dormant cells that survive antibiotic treatment [45,51].

Research on the druggability of FtsZ requires suitable methodology to identify drug candidates and to determine their mechanisms of action. Decades of research into how FtsZ functions in cell division have provided numerous approaches to assess its self-association and its direct or indirect interaction with protein partners, nucleic acids, or the membrane that may be adapted for this purpose. Simple assays amenable to high throughput screening are useful for initial identification of lead compounds. Further elucidation of mechanisms, required for a drug to be approved [52,53], would benefit from orthogonal approaches [54] and reconstruction in cell-like systems to evaluate the effects of crowding, compartmentalization, and the membrane surface on these mechanisms [55,56,57,58]. Such assay systems are particularly useful in cases where the targeted protein complexes mainly assemble in crowded conditions only found inside the cell and/or on lipid membranes.

Reflecting the growing interest in FtsZ as a target for the development of new antibiotics, comprehensive reviews have been very recently published dealing with descriptions of the families of candidates already identified and their mechanisms of action, mainly involving the alteration of FtsZ self-association to form GTP induced polymers [1,4,59,60,61]. Consequently, here we will focus on areas less covered, including the assays that have been used, or are potentially useful, to identify new anti-FtsZ compounds. We will discuss how such assays can help evaluate the molecular mechanisms of drug candidates in dilute solution and through biochemical reconstruction in minimal membrane systems. We will also highlight the importance of the self-association of FtsZ in the GDP bound state and its interactions with division regulators in crowding conditions leading to higher order membraneless structures, possibly related with persistent states and hence of interest for further understanding of antibiotic resistance mechanisms.

## 2. Detection and Quantification of Direct FtsZ-Drug Binding in Solution

Identification of novel antimicrobials directed towards FtsZ requires suitable methods to evaluate direct drug-protein interactions (Figure 3, Table 1). Moreover, binding affinity measurements of the identified candidates are a necessary starting point for the determination of minimum inhibitory concentrations (MICs). One of the methods that have been employed for this purpose is fluorescence anisotropy. Anisotropy is very sensitive to size changes, such as those that occur when a small fluorescent species binds to a large macromolecule [62,63]. Among the most widely employed techniques for drug screening [64], fluorescence anisotropy can precisely measure high affinity interactions, and several low volume samples can be simultaneously and rapidly analyzed by using a plate reader equipped with polarizers. Anisotropy has been used to determine the binding affinity between a boron-dipyrromethene (BODIPY)-labeled oxazole-benzamide inhibitor and FtsZs from a variety of Gram-negative and Gram-positive bacteria [65]. In addition, this technique has been used to determine interactions between FtsZ and 4′,6-Diamidino-2-phenylindole (DAPI), a DNA intercalating dye that is also a modulator of FtsZ assembly [66], and the guanidinomethyl biaryl compound 13, a broad spectrum bactericidal agent [67]. An anisotropy-based competition assay has been devised to identify compounds targeting the GTP binding site of FtsZ by using the fluorescent analog 2′/3′-O-(N-methylanthraniloyl) (*mant*)-GTP. This assay has been used to analyze the binding of compounds such as C8-substituted GTP analogs to FtsZ from the archaeon *Methanococcus jannaschii,* or the interaction of a variety of synthetic inhibitors of bacterial cell division with *Bacillus subtilis* FtsZ [68,69]. Insight into the mechanism of action of the above mentioned compound 13 has also been obtained through anisotropy measurements of BODIPY-labeled GTPγS, showing competition with the drug for binding to the GTP site of FtsZ [67].

A reference technique for the thermodynamic characterization of binding reactions is isothermal titration calorimetry (ITC), based on the measurement of the heat change produced upon interaction [70]. This technique is therefore ideal for detection of inhibitory compounds and selection of candidates showing optimal affinities for their targets. Binding isotherms of the inhibitory MciZ peptide to FtsZs from *B. subtilis* or *Staphylococcus aureus* demonstrated that the peptide, produced by the former during sporulation to arrest cytokinesis by disrupting FtsZ polymerization, binds FtsZ from different bacteria by establishing entropically favorable interactions [71]. This method also allowed determination of the binding constants of MciZ mutants with FtsZ. Likewise, ITC was employed to characterize the binding of *E. coli* FtsZ to cinnamaldehyde, a plant-based small molecule that inhibits FtsZ polymerization and GTPase activity [72].

As for other targets, the identification and characterization of hit compounds acting on FtsZ has benefited from the use of bioinformatics tools and structural approaches like nuclear magnetic resonance spectroscopy (NMR) and X-ray crystallography [48]. With these methods, the precise binding sites of the candidates can be ascertained and chemical modifications improving affinity or specificity can be foreseen by docking onto the FtsZ structures available. The use of methods such as NMR and X-ray crystallography to assess direct binding of inhibitors to FtsZ has been thoroughly analyzed elsewhere [4].

## 3. Methods to Identify Drugs Targeting FtsZ Polymerization and Their Mechanisms

Given the crucial role of GTP-induced polymerization of FtsZ in cell division, alteration of this process by chemicals that either block assembly or prevent disassembly has been so far the most pursued strategy to perturb the function of this protein. Methods to measure FtsZ polymerization are of great utility in the identification of such compounds and in the determination of their precise mechanisms of action and active concentrations (Figure 3, Table 1). 90° Light scattering (LS), which can detect polymer formation, monitor time-dependent polymer disassembly, and determine the critical concentration for polymerization [73], is probably the most widely employed method to this end. This technique has been used to investigate how DAPI inhibits FtsZ polymer dynamics, how compound 13 [67] stimulates FtsZ polymerization in a concentration-dependent manner, or how the aminobenzamide derivative compound 28 [74] inhibits assembly of FtsZ from *S. aureus* [4,66,67,74]. GTP-induced FtsZ polymers can also be detected by centrifugation, followed by evaluation of the fraction of protein in the pellet and supernatant, for example by polyacrylamide gel electrophoresis. This approach has been employed to study the impact of the polyhydroxy aromatic compound UCM05 [69] and the benzamide antibiotic compound 8j [75] on the polymerization of FtsZ from *B. subtilis* [69,75] and of bacteriophage λ Kil peptide (Figure 1c) on the assembly of *E. coli* FtsZ filament bundles [76]. Procedures for the determination of the GTPase activity of the protein (reviewed in [48]) are frequently employed in studies of its druggability (see for instance [77]).

Another method for the characterization of FtsZ polymers is dynamic light scattering (DLS). The autocorrelation functions obtained by DLS result from the random diffusive motion of macromolecules in solution that correlate with their mass and shape, which translates into fluctuations in scattered light. Analysis of these curves yields translational diffusion coefficients [78] that can reflect changes in mass, for example from dissociation of protein-protein interactions upon drug binding, as long as there is a mass difference of at least four-fold. Information about the polymerization/depolymerization kinetics can be obtained by following the intensity of scattered light with time. Commercial plate readers are available that make DLS measurements amenable for high throughput screening of compounds [79]. DLS was used to determine the absence of any stabilizing effect of the benzamide derivative PC190723 in FtsZ filaments of *Caulobacter crescentus* [80]. This is consistent with the idea of this type of small molecule binding only weakly to FtsZ from Gram-negative bacteria [5], as previously suggested by inhibition of cell division in *S. aureus* and *B. subtilis* but not *E. coli* [81]. In a different approach, DLS was used to test the activity of Temporin L, an antimicrobial peptide active against Gram-positive and Gram-negative bacteria, on *E. coli* cells [82]. Specifically, DLS data revealed the formation of elongated structures by the inability of the cells to divide, which arose from the interaction of the peptide with FtsZ as determined by parallel fluorescence and GTPase activity measurements. A complementary technique, ITC, was used in an unconventional application to discern the different assembly mechanisms governing FtsZ oligomerization and polymerization [83]. It would be possible in principle to characterize the effect of compounds that interfere with these assemblies, although this would require the application of complex mechanistic models to analyze the data.

Tunable resistive pulse sensing (TRPS) is an emerging sizing method applicable to a wide variety of particles ranging from single molecules to whole cells within 40 nm–10 μm in size [84]. It measures current fluctuations produced by the species passing through a dynamically resizable pore, allowing simultaneous determination of size and concentration of the species present in polydisperse samples [85]. TRPS is a highly accurate and precise method potentially useful to detect and quantitate the effect of drugs on FtsZ polymerization.

FtsZ polymerization has also been studied by fluorescence correlation spectroscopy (FCS), a single molecule technique that obtains autocorrelation curves from fluctuations in fluorescence intensities, which can then be used to derive translational diffusion parameters of the fluorescent molecules [63,86]. FCS methods are suitable for high throughput screening [87]. An orthogonal combination of FCS, DLS, and sedimentation velocity (SV) approaches has been used to study FtsZ polymers [54]. Sedimentation velocity experiments yield sedimentation coefficients that correlate with the sizes and shapes of the molecular species, and parallel determination of translational diffusion coefficients by DLS or FCS allows calculation of masses using the Svedberg equation [54]. FCS and SV have been jointly applied to characterize the inhibition of FtsZ polymerization by Kil peptide when it is expressed from infecting λ phage (Figure 1c), with potential application in the development of therapies against bacterial infections [41]. That study also confirmed direct interaction between Kil and FtsZ by fluorescence anisotropy. Fluorescence cross-correlation spectroscopy (FCCS) is an extension of FCS [63,88,89] and relies on two-channel detection for the generation of cross-correlation curves indicative of complex formation. FCCS was used in high throughput screens to identify compounds that perturb FtsZ self-interactions [90]. In this assay, N- and C-terminal fragments of FtsZ were each labeled with fluorescent proteins emitting at different wavelengths, and hits were based on the ability to reduce the cross-correlation arising from GTP-induced interaction between the fragments [90]. Procedures based on fluorescence methods such as Förster resonance energy transfer (FRET) or anisotropy for the measurement of the critical concentration of polymerization of FtsZ are also potentially applicable to the screening of drugs that interfere with FtsZ assembly [30,91].

The biophysical and biochemical methods described above are usually complemented with imaging approaches that provide structural information, the most popular being electron microscopy. This technique has been used to detect FtsZ polymer bundling induced by drugs such as compound 13 or DAPI, or morphological changes in FtsZ polymers induced, for example, by compound 8j [66,67,75]. The use of conventional fluorescence imaging methods for this purpose is more limited, as their lower resolution precludes characterization of the small polymers FtsZ forms in dilute solution conditions. They may be used, however, for studies of the modification of FtsZ bundles observed in crowding conditions in vitro [31] by antibacterial compounds and they are one of the preferred tools for the in vivo investigation of drug mechanisms of action.

## 4. Exploiting the Interactions of FtsZ with Binding Partners in Solution to Discover New Antimicrobials

Relatively unexplored targets to arrest bacterial cell division with therapeutics are the multiplicity of interactions that FtsZ establishes with a variety of proteins, some of them conserved across multiple bacterial species (Figure 1). There is agreement, however, that hetero-associations of FtsZ would constitute excellent targets for the identification of anti-infectives, probably with lower propensity to develop resistance [1,50]. In addition to the opportunities for the identification of broad-spectrum antibacterials acting on FtsZ, targeting its hetero-associations may help in the quest for antibiotics directed towards certain bacterial species where those partners specifically operate. The identification of antibacterial candidates targeting heterotypic interactions within the FtsZ network will benefit from the many assays developed to detect and quantify these interactions in solution and in reconstructed systems (see Section 5).

Probably the most widely explored interaction of FtsZ with the purpose of finding antibiotic leads is with ZipA (Figure 1b), a protein specific to enterobacteria, which anchors the Z-ring to the membrane [18]. Together with FtsZ and FtsA, in *E. coli*, ZipA forms the proto-ring, a protein complex that subsequently recruits other components of the divisome [20]. Since ZipA is a membrane protein whose N-terminal region is actually inserted in the bacterial inner membrane, assays in solution are usually conducted with mutants lacking this region but which keep the C-terminal globular domain recognized by FtsZ. The full-length ZipA protein has been also employed in experiments using cytomimetic membranes for its reconstruction and/or solubilization (see Section 5). Years ago, a fluorescence anisotropy assay was used to screen drug candidates that perturb the essential FtsZ-ZipA interaction in *E. coli* [92]. This assay consisted of a fluorescently labeled peptide derived from the C-terminal tail of FtsZ that interacts with ZipA, together with the cytoplasmic C-terminal domain of ZipA that participates in this interaction. NMR was also used for the screening of compounds that bind to this C-terminal region of ZipA [120].

Another well-studied case of hetero-association is that of FtsZ with the bacterial protease Clp (Figure 1d), also considered a novel target for antibiotic action because of its implication in vital cellular functions [121]. Clp is found in most bacterial species and involved in regulatory proteolysis. In *E. coli*, the proteolytic complex is formed by ClpP that includes the proteolytic core, and the unfoldase ClpX that controls access of the protein substrate to that core [21]. Antibiotic acyldepsipeptides (ADEPs) have been described to induce rapid degradation of FtsZ in vivo and in vitro [1,122] by deregulating the function of Clp [123]. Degradation studies with *B. subtilis* FtsZ have identified the disordered N-terminus and, at high concentrations of ADEP2, also the C-terminus of FtsZ as the targets for the ADEP-ClpP complex, and suggested it leads to depletion of the FtsZ cytoplasmic pool under certain conditions [121]. Along these lines, treadmilling within early Z-rings seems to be more affected by FtsZ depletion, as determined by time-lapse fluorescence and super-resolution microscopy experiments [98]. A number of these ADEP compounds proved to have significant antimicrobial activity, mostly in Gram-positive bacteria, and have resistance-breaking potential, thus highlighting the inhibition of cell division as a powerful strategy for the development of new antibiotics [1].

An attractive interaction for the identification of broad-spectrum antibacterials is that of FtsZ with FtsA (Figure 1b), the other membrane anchor of the division ring, more widely conserved than ZipA. FtsA is an amphitropic protein that interacts with the membrane through its C-terminal amphipathic helix and with FtsZ through its core conserved domain [124]. FtsA interacts also with numerous divisome proteins, being directly or indirectly involved in septal wall and peptidoglycan synthesis [2]. The interaction between FtsA and FtsZ from *E. coli* decreased FtsZ’s GTP hydrolysis [125] although conflicting results have been obtained in this regard with FtsAs from different microorganisms, possibly related to species-specific regulatory mechanisms [2]. The binding affinity of FtsZ-FtsA from *Vibrio cholerae* has been measured by following the variation of the intrinsic tryptophan fluorescence of FtsA upon interaction with FtsZ, which lacks tryptophan residues [94]. As FtsA is an amphitropic protein, minimal membrane systems have been also used to reconstruct its interactions (see Section 5).

The formation of the Z-ring at the correct time and place largely depends on the spatial and temporal regulation of FtsZ assembly/disassembly dynamics by agonists and antagonists, mostly exerted through the direct binding of protein factors to FtsZ (Figure 1). Chemicals altering these interactions would therefore constitute potential antimicrobial leads. Broad-spectrum antibiotics could be obtained by targeting the interactions of FtsZ with ZapA (Figure 1e), a widely conserved protein that positively regulates Z-ring formation, by linking FtsZ with the chromosomal replication terminus region (Ter macrodomain) in the case of *E. coli* [22]. Fluorescent chimeras have been employed to show delocalization of this ZapA from midcell upon treatment of *B. subtilis* cells with the benzimidazole derivative BT-benzo-29 [99] or colocalization with FtsZ foci dispersed all over the cytoplasm, induced by compound 8j [75].

Among the negative regulators of FtsZ assembly, the nucleoid occlusion factor SlmA from *E. coli* and the Min system have been widely studied in order to unravel their mechanisms of action. SlmA binds to specific DNA sequences (SBS) found along the bacterial chromosome except in the Ter macrodomain, blocking inappropriate Z-ring assembly over un-partitioned chromosomes through direct interaction with FtsZ ([126], Figure 1a). The MinCDE complex blocks FtsZ polymerization at the cell poles through pole-to-pole oscillatory waves along the cytoplasmic membrane (Figure 1f), where MinC directly interacts with FtsZ [9]. Multiple biochemical and biophysical assays have been devised to measure the interactions of FtsZ with these antagonists that may be used to identify drug candidates. For example, anisotropy-based assays have been employed to measure the binding of FtsZ to MinC or to SlmA nucleoprotein complexes [38,39,93]. In addition, biosensor assays with immobilized proteins have been used to measure the interaction of FtsZ with MinC variants or of FtsZ variants with SlmA [12,96], and the binding of FtsZ to MinC has been detected by FRET [95]. The interactions of SlmA with FtsZ polymers and with the FtsZ-GDP oligomers have been assessed by cosedimentation and analytical ultracentrifugation, respectively [93,127]. Finally, ITC has been used to measure the binding of FtsZ from *E. coli* to SulA protein (Figure 1d), an inhibitor that delays division as part of the SOS response upon detection of damage to chromosomal DNA [40].

## 5. Reconstruction of Cellular FtsZ Subsystems

Reconstruction of selected subsets of proteins within the FtsZ interaction network in synthetic platforms that display different features of the cellular environment, relevant for the function of this protein, may be a suitable strategy for the identification of antimicrobial candidates. Not only can these approaches shed further light onto mechanisms used by molecules identified through assays in dilute solutions or in cells, but also the measured effects of these molecules would more closely resemble those occurring in vivo, potentially facilitating the selection of candidates. One of the key elements to be considered in such studies is the cytoplasmic membrane, which plays an active part in divisome function, as most essential divisome proteins interact directly with it. Some divisome proteins anchor FtsZ polymers to the membrane (FtsA, ZipA; Figure 1b), some directly interact with the membrane surface at some point while exerting their function (MinDE, SlmA, MatP; Figure 1a,e,f), and others are integral membrane proteins (FtsK, FtsEX, the FtsQBL complex, and the FtsWI septum synthesis enzymes). Consequently, it is important to include membranes in order to evaluate more accurately the interactions of FtsZ and to study protein-membrane interactions that regulate cell division. The different available approaches start with protein-free lipid bilayers, to which proteins are externally added (Figure 3, Table 1). For a detailed review of membrane transformation by molecular assemblies, see [57].

Membrane proteins can be stabilized in solution by reconstruction in phospholipid bilayer nanodiscs, which are small portions of a lipid bilayer surrounded by two copies of an amphipathic scaffold protein [100]. Nanodiscs enable the analysis of membrane protein interactions by the many techniques available to study soluble proteins, in a more native-like environment than when using detergents. Nanodisc technology has been applied to the analysis of interactions of full-length (including the transmembrane region) ZipA (Figure 1b) with FtsZ oligomers and with GTP-induced FtsZ polymers [37]. The ZipA-interacting C-terminal tail of FtsZ was used in that study to prove the utility of analytical ultracentrifugation and FCS assays developed to find compounds that interfere with the FtsZ-ZipA interactions.

Reconstruction of membrane protein interactions have also been conducted using microbeads of various materials, such as silica or polystyrene, and of uniform size, coated with a lipid bilayer or monolayer. Such lipid-coated microbeads have been used to characterize the binding of FtsZ oligomers and polymers to a His-tagged ZipA variant anchored to the lipid membrane, as well as the influence of receptor density on this binding, by measurement of the fluorescence signal corresponding to the amount of unbound protein in the supernatant after sedimentation of the microbeads [101]. The same procedure was used earlier to quantify the interaction of FtsA (Figure 1b) with membranes of different compositions, including the *E. coli* cytoplasmic membrane [102]. The affinity of FtsA to membranes was highly dependent on lipid composition, indicating that this approach could serve as a basis for the characterization of the effects of compounds on membrane remodeling [125] and/or FtsZ anchoring, both modulated by FtsA during cytokinesis. Experiments with coated microbeads also provided evidence that SlmA (Figure 1a) can interact with lipids and, only upon FtsZ interaction, recruit its targeted specific DNA sequences to these lipids [103]. Although more laborious than other types of experiments, this procedure is highly reproducible and suitable for the screening of compounds that affect these interactions, as tested by proof-of-concept experiments with molecular inhibitors of the FtsZ-ZipA interaction [128].

Supported lipid bilayers (SLBs), mostly in combination with high-resolution fluorescence microscopy imaging, have been extensively used to characterize the bacterial division machinery and in particular FtsZ. Initial approaches used a fusion of FtsZ to a fluorescent protein and a membrane targeting sequence (FtsZ-YFP-mts), thus overcoming the need of an anchor protein. Reconstitution experiments with these flat bilayers showed the effect of spatial patterning by the MinCDE complex (Figure 1f) on the distribution of FtsZ [104]. SLBs also served as platforms to demonstrate the self-organization of FtsZ polymers into dynamic ring patterns on their own [129] or with FtsA [105]; to show treadmilling of individual FtsZ filaments [105]; to highlight the possible role of ZipA as passive membrane anchor for FtsZ filaments [106]; and, finally, to report the comigration of other divisome proteins with treadmilling FtsZ filaments [107]. Very recently, SLBs were elegantly used as a platform to generate dynamic planar waves of de novo-synthesized MinDE proteins, which spatially regulate FtsZ filament patterns in the presence of de novo-synthesized FtsA [108]. This cell-free expression system allows protein production from a DNA template through the PURE system [130], a mimic of the molecular composition of the bacterial cytoplasm, but which lacks endogenous proteins that might interfere with the reconstituted system. Towards the goal of identifying new antimicrobials, SLBs have been mainly used to characterize the effect of antimicrobial peptides (AMPs) on the membrane, such as formation of lipid fibrils with Temporins B and L [131], or membrane pore formation by the AMP model amhelin [132].

Biosensors can be used for the real-time quantitative characterization of protein-protein and protein-membrane interactions. Biolayer interferometry measures the change in the interferometry signal upon protein interaction after immersion of a biosensor tip, coated with a supported bilayer, into a solution with the protein. This technique has been used, in parallel with coated microbeads, to determine SlmA (Figure 1a) binding to a lipid membrane [103]. These kinds of measurements can be easily employed in the detection of inhibitors of the interaction of SlmA with FtsZ and/or the lipid membrane, potentially interfering with the proper localization of the Z-ring to the midcell division site. The ability to rapidly measure very small volume samples makes this method suitable for compound screening. NanoSPR (surface plasmon resonance) measures the changes in refractive index produced upon binding of a molecule to gold nanorods, and has been used to measure local concentration of the Min proteins (Figure 1f) resulting from their oscillations, i.e., the intrinsic dynamics of the system, using nanorods coated with *E. coli* lipids [111]. As in the case of nanodiscs, the C-terminal FtsZ peptide has been used to validate assays potentially useful to identify inhibitors of FtsZ-ZipA interactions, based on lipid-coated plasmonic nanosensors with an immobilized soluble version of ZipA [109] (Figure 1b). Finally, plasmonic sensors based on surface-enhanced Raman scattering have been used to detect the interactions between FtsZ and bacterial membrane elements [110]. From this, fast assays to screen for drugs altering the attachment of FtsZ to the membrane could be developed.

A further step towards reproducing true cellular macromolecular interactions in vitro consists of reconstruction in biomimetic environments, which reproduce membrane boundary and confinement conditions similar to those found in vivo [55,56,57,58]. This approach allows characterization of a particular cell function in terms of the minimal number of biological components involved in the interaction pattern (including the membrane as an active element), their dynamics, and their localization within the cell-like environment. Importantly, studies of potential inhibitors in crowded environments, although technically more complex, could more reliably assess their effect. Intracellular crowding is known to greatly affect the reactivity of molecules and enhance their associated states through volume exclusion, in turn modulated by other nonspecific attractive and repulsive effects [133,134,135]. It is therefore probable that functional complexes resembling their physiological counterparts form in vitro only under those conditions, which would contribute valuable information on a drug candidate’s mechanism of action and dose-response as opposed to assays conducted in diluted conditions. A plethora of different approximations exists that, although not yet extensively used for identifying inhibitors, hold great promise as platforms for screening and characterization of compounds that inhibit bacterial cell division (Figure 3, Table 1). Among them, microdroplets (inverted micelles) and liposomes, particularly giant unilamellar vesicles (GUVs), have been employed most often because of their membrane fluidity and size range similar to bacterial cells. These sizes are ideal for the characterization of mesoscale cytoskeletal systems, allowing visualization of polymer networks formed by FtsZ (below), or the bacterial actin homolog MreB [136]. Similar systems have been used to monitor polymers of eukaryotic tubulin [137], actin [138], and myosin [139] with an obvious application for drug testing.

In a notable use of liposomes to address whether FtsZ polymers can exert a constrictive force on the cytoplasmic membrane, chimeras of FtsZ fused to both YFP and an artificial membrane targeting sequence (mts, mentioned above) were externally added to multilamellar liposomes. These chimeras exerted different forces on the liposome surface depending on the position of the mts sequence, with either concave depressions and tubulation (FtsZ-YFP-mts) or convex bulges (mts-FtsZ-YFP) [140]. Similar FtsZ chimeras were added to tubular lipids to characterize the arrangement of FtsZ filaments from *E. coli* and *Mycobacterium tuberculosis* on their surface [141]. The de novo-synthesized MinDE system (Figure 1f), mentioned as part of the SLB reconstructions, has been shown to oscillate between membrane and lumen and produce autonomous deformation of liposomes [108]. The same cell-free gene expression system was employed in liposomes and lipid bilayers to reconstitute FtsZ-FtsA ring-like structures that were able to constrict the liposomes [113].

Reconstitution of FtsZ and a ZipA (Figure 1b) variant into permeable GUVs resulted in membrane shrinkage modulated by dynamics of FtsZ polymers, which were externally triggered by diffusion of GTP and magnesium through membrane pores. Vesicle collapse was inhibited by addition of the FtsZ C-terminal peptide, which interfered with FtsZ-ZipA interaction [114]. Deformation of the membrane was also achieved by de novo synthesis of FtsZ and ZipA in GUVs, but not with the FtsZ-mts chimera [115]. Finally, lipid vesicles produced by manual emulsion were deformed into rod-shaped containers by trapping them in microfluidic devices, allowing visualization studies of FtsZ-mts filament arrangement in bacterial cell-like geometries [142].

Cell-like systems with precisely controlled composition, such as microdroplets generated by microfluidics [112], provide avenues for quantitative analysis of proteins in a crowded environment and confined by a lipid boundary. This technology allowed the measurement of self-organizing behavior of FtsZ polymers [117] and its modulation in a system that mimics a compartmentalized cytoplasm [118]. The use of traps in microfluidics chips provided a way to assess the effects of microdroplet shape and size on the arrangement of FtsZ polymer networks in the presence of ZipA (Figure 1b) [116]. Microdroplets containing FtsZ could also be transformed off-chip into permeable GUVs by an adapted droplet transfer method [118]. This strategy resulted in a lipid bilayer more closely resembling the membrane, where protein polymerization could be triggered from the outside. In this way it was shown that FtsZ could relocate within compartments, depending on its self-association state [118], as previously determined upon depolymerization of FtsZ in microdroplets generated by manual emulsion [119]. In addition, direct generation of GUVs in the microfluidic chip, which involves the use of organic solvents [143,144], has been used to measure the sequestration of FtsZ by coacervates [145]. Because each generated microdroplet/GUV is identical and they are produced in very large numbers with low sample consumption, the use of microfluidics facilitates high throughput screening of compounds [112,146].

Finally, microfluidic platforms with very diverse designs have been used to test the activity of inhibitory compounds in miniaturized bacterial cell cultures, both in systems where bacteria encounter gradients of the inhibitors within the microfluidic channels, or where bacteria are encapsulated together with the inhibitors inside microdroplets (thoroughly reviewed in [147]). The latter approach has substantially higher technical requirements, limiting its use for bacterial analysis. The low volume reactions and multiplexing capacity permit high throughput assays that can test many potential inhibitors for a particular bacterial species as well as identify antibiotic resistance.

## 6. Could FtsZ Biomolecular Condensates Help Understand Persisters?

Bacterial antibiotic resistance is caused by many factors, including degradation of the antibiotic or modification of the antibiotic target. Resistance to treatment can be thus genetically acquired, but it can also be achieved through persisters, defined as a fraction of the total cell population that is transiently tolerant to antibiotics [51]. These persister cells often display slow or even arrested growth, which resumes after the end of the treatment [148]. The molecular mechanisms underlying persister formation and regrowth have remained elusive. Recently, connections have been established between condensate formation by phase separation events in the cytoplasm and the presence of these dormant or persister cells in bacterial populations [45]. This is consistent with evidence showing the formation of membraneless compartments in different kinds of cells to adapt to stress conditions [149].

Indeed, FtsZ purified from *E. coli* forms biomolecular condensates together with one of its binding partners, the DNA binding protein SlmA (Figure 1a), in crowding conditions in bulk or encapsulated inside cell-like systems [42]. Phase separation could be caused by the multivalent interactions established by these proteins, enhanced by crowding and by the presence of the DNA sequences targeted by SlmA. In the absence of SlmA, FtsZ by itself can achieve a sufficiently dense network of interactions to phase separate, but only at high magnesium and low potassium concentrations (Figure 2) [43]. Both the homotypic and heterotypic FtsZ condensates described so far involve the GDP form of the protein, which is still active for polymerization when GTP is added to the condensates. A putative role of condensates in the regulation of FtsZ spatial distribution and function, and ultimately in the mechanism leading to division in normal bacteria, has been suggested [42,43]. It has also been hypothesized that the reversible formation of these condensates could respond to stress signals. One way in which these signals could regulate FtsZ biomolecular condensation is through changes in the relative abundance of guanine nucleotides (Figure 4). Indeed, persister cell formation usually depends on (p)ppGpp signaling [150], which promotes bacterial survival by activating global metabolic signals that downregulate the activities of enzymes involved in GTP biosynthesis [151,152]. Based on the in vitro data, reduction in cellular GTP levels would be expected to shift the condensate/filament equilibrium towards condensate assembly, thus reducing the amount of FtsZ available for Z-ring formation needed to drive cytokinesis (Figure 4).

The dynamism and reversibility of FtsZ biomolecular condensates would also explain how persisters resume growth after the stress is over: once normal GTP levels are restored, the protein within the condensates will still be able to assemble into FtsZ polymers and hence into a functional Z-ring (Figure 4). This hypothesis is supported by the recent identification of a novel subcellular membraneless structure in non-growing late stationary phase bacteria, termed a regrowth delay body, linked to multidrug tolerance in *E. coli*, *Salmonella typhimurium* and *Shigella flexneri* [153]. Yu and coworkers found that FtsZ accumulates in these bodies, presumably in a folded conformation, and that under regrowth conditions these bodies disband and Z-rings reassemble at midcell. Interestingly, the extent of regrowth delay body formation correlated with the degree of antibiotic tolerance. Moreover, a combination of rifampicin with ADEP4 was able to eradicate persister cells in a chronic biofilm infection through a nonspecific protease activity of ClpP that strongly degraded FtsZ, among many other proteins [154].

FtsZ biomolecular condensates may therefore constitute additional targets to be considered in the search for new antimicrobials, as it has been recently proposed for condensates involved in different diseases [155]. Drug candidates that modulate the formation of these structures may alter cell functionality and, more importantly, they may lead to novel strategies to combat persistence, a major health threat. The assays so far developed to study FtsZ biomolecular condensates would help to identify such candidates. Confocal fluorescence microscopy imaging has been an invaluable tool for the identification and characterization of FtsZ biomolecular condensates in crowding solutions in bulk and in cytomimetic systems [42,43]. In those studies, confocal microscopy measured the size and shape of condensates formed by the scaffold biomolecules. Time-lapse confocal imaging monitored the conversion of FtsZ condensates to GTP-induced filaments and back, as well as the capture of client molecules from colocalization analysis, confirming their dynamism. Those studies also measured turbidity over time, which provided more quantitative information about the factors affecting condensate formation and permitted simultaneous evaluation of dozens of samples using a plate reader. Parallel measurements by confocal imaging and turbidity provided valuable information about how crowding, protein and nucleic acid concentration, salt and divalent cations fine tune condensate assembly by FtsZ [42,43].

Reconstruction in lipid-stabilized microdroplets was also conducted in the aforementioned studies to get some insight into the spatial distribution of the FtsZ condensates in a confined and compartmentalized cell-like environment. Synthetic biology approaches are very promising for the investigation of biomolecular condensates involving bacterial factors because the visualization and tracking of condensates in cells as small as bacteria pose a significant challenge. The described methodologies provide a good starting point for the screening of molecules that deregulate condensate assembly, block evolution towards filaments in response to nucleotide, or modulate FtsZ condensate dynamics and spatial distribution in cell-like systems. The growing number of methodologies that are currently being applied or adapted to the study of biomolecular condensates [156] provide an excellent opportunity to devise additional assays for these FtsZ assemblies. Among the techniques previously applied to study FtsZ interactions, FCS [157] and DLS [158] have been recently used to analyze biomolecular condensates. Other research directions towards the possible exploitation of FtsZ condensates as targets would involve whether they can be promoted by cell division proteins other than SlmA and the identification of condensates formed by FtsZ from other species.

## 7. Conclusions

The search for new antibiotics active in bacterial division has mainly focused on FtsZ, because of its ubiquity and essential central role, and specifically on the disruption of FtsZ polymerization. However, alternative interactions are worth exploring. Although GTP dependent polymerization of this protein is crucial to accomplish cell division, interactions with a wide variety of partners are in most cases indispensable for bacterial survival. Hence, the interactions of FtsZ with other divisome elements, some of them acting at different levels during division, could be promising targets, and their number will likely increase as their functions are better characterized. In addition, the traditionally overlooked self-association of FtsZ-GDP is emerging as a putative target, given its role in the assembly of biomolecular condensates. Considering the implications of these types of higher order dynamic assemblies in a variety of diseases, including infections, it would not be surprising that this form of the protein attracts more attention in the field.

There is a panoply of robust established techniques commonly used for the characterization of macromolecular interactions that can be easily employed for the assessment of the activity of a drug with its target. Drug development, and likewise evaluation of drug resistance, will greatly benefit from emerging and/or adaptation of available methodologies to evaluate drug activity in biologically relevant environments. Studies in reconstructed systems will likely provide more reliable data on the screening of compounds. For example, drug candidates previously discarded because of low affinity might actually exhibit better activities under cell-like conditions, and similarly drugs with high affinity interactions in non-physiological purified systems may turn out to be less effective in the cell environment. Consequently, we predict that the use of more cell-like systems for screening assays will result in better drug candidates in the future.

## Figures and Tables

**Figure 1 antibiotics-10-00254-f001:**
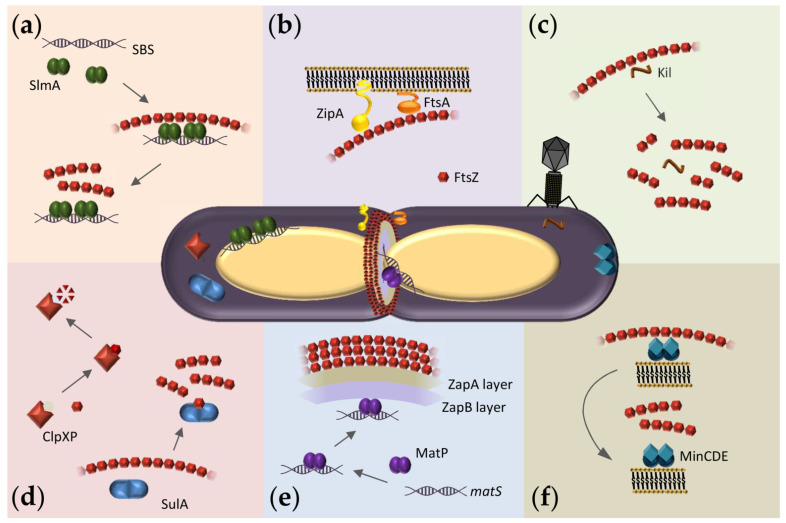
Overview of FtsZ hetero-associations in *E. coli* as possible targets for antibacterials. At the center, representation of a dividing *E. coli* cell showing the Z-ring assembled at midcell and FtsZ macromolecular interaction systems. (**a**) Scheme of the nucleoid occlusion system mediated by SlmA. SlmA bound to specific SBS sequences within the bacterial chromosome, except in the Ter macrodomain, inhibits FtsZ polymerization, thus protecting the chromosome from aberrant scission. (**b**) The Z-ring is attached to the inner membrane through the protein anchors ZipA and FtsA. (**c**) Interaction of FtsZ with the bacteriophage λ Kil peptide disrupts FtsZ polymers, resulting in shorter oligomers of variable size. (**d**) SulA protein, induced by the SOS response, transiently inhibits cell division during DNA repair by sequestering FtsZ monomers. ClpXP protease degrades FtsZ, modulating polymer dynamics and Z-ring formation. (**e**) Scheme of the Ter- linkage that contributes to Z-ring positioning at midcell. From the membrane inwards, FtsZ interacts with ZapA, ZapB, and MatP. MatP is bound to *matS* sequences located in the chromosomal Ter macrodomain. (**f**) The pole-to-pole oscillatory waves of the MinCDE system on the cytoplasmic membrane prevent FtsZ polymerization at the cell poles.

**Figure 2 antibiotics-10-00254-f002:**
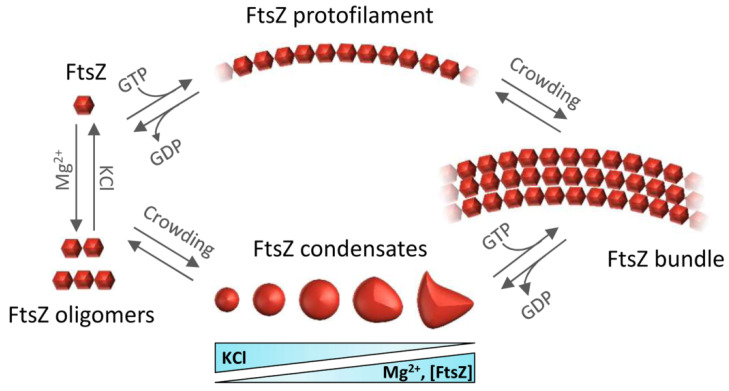
Overview of the assemblies formed by FtsZ self-association. FtsZ-GDP monomers form oligomers favored by magnesium and low salt. Under crowding conditions, FtsZ-GDP reversibly phase-separates into biomolecular condensates whose number and size depend on FtsZ, KCl, and Mg^2+^ concentrations. GTP addition triggers FtsZ polymerization whether the protein is in oligomers or in condensates. Macromolecular crowding favors the lateral association of single stranded filaments to form bundles.

**Figure 3 antibiotics-10-00254-f003:**
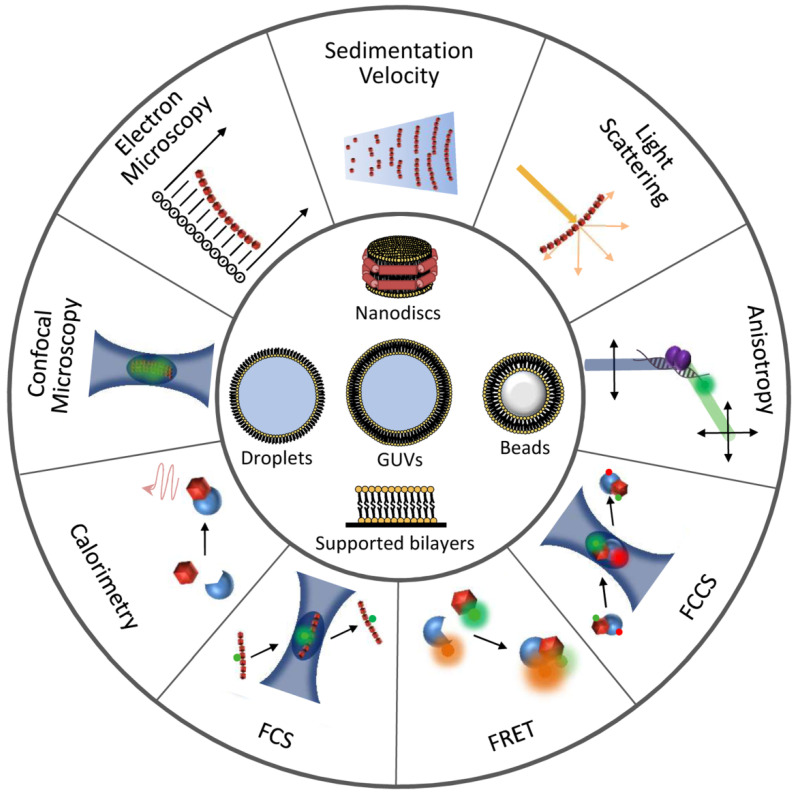
Illustration of techniques and reconstruction approaches useful for the detection and characterization of FtsZ interactions. Experimental approaches to study FtsZ homo- and hetero-association are represented at the outer circumference. Minimal membrane systems and cytomimetic platforms are shown at the center.

**Figure 4 antibiotics-10-00254-f004:**
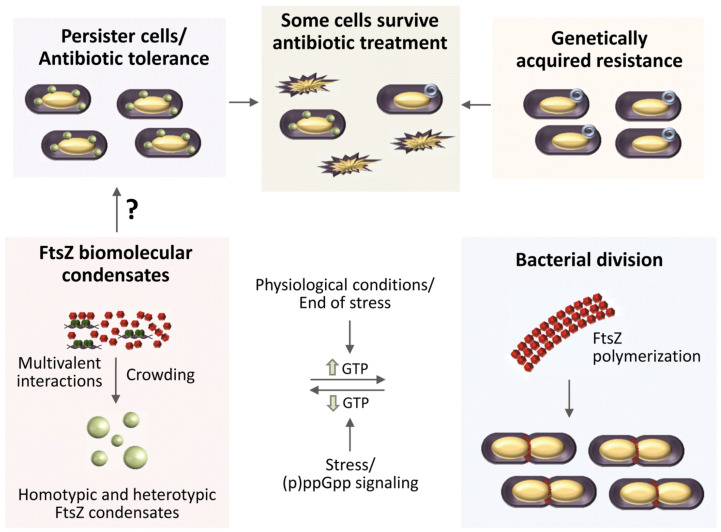
Hypothetical role of FtsZ biomolecular condensation in bacterial resistance to antibiotic treatment. Bacteria can survive antibiotics through a subpopulation of transiently tolerant persister cells or by genetically acquired resistance. The mechanisms leading to formation of persisters remain unclear, but the biomolecular condensates formed by FtsZ, by itself or with partners, may play a role. Homotypic and heterotypic phase-separated condensates of FtsZ-GDP are favored by multivalent interactions and crowding in cells. Condensates would be also favored in situations with low GTP levels, such as stress conditions leading to the activation of (p)ppGpp signaling after antibiotic exposure. The reversible conversion of FtsZ condensates into FtsZ polymers, triggered by GTP, would explain how the persister cells resume growth after stress conditions are over and normal GTP levels are restored.

**Table 1 antibiotics-10-00254-t001:** Biophysical and reconstruction methods useful for the characterization of FtsZ homo-, hetero-associations, and interactions with drugs. Examples marked with an asterisk are mentioned to illustrate the potential use of the technique.

Method	Fundamentals Refs.	Information Obtained	Examples Refs.
**Techniques in solution**			
90° LS, Sedimentation, EM	[54]	Assessment of polymerization	[4,66,67,69,74,75,76]
Fluorescence anisotropy	[62,63]	Quantification of drug binding	[65,66,67,68,69]
		Assessment of polymerization	[91] *
		Interaction with Kil, ZipA, MinC, SlmA	[41,92], [38,39,93] *
FCS, FCCS	[63,88]	Assessment of polymerization	[41,90]
ITC	[70]	Quantification of drug binding	[71,72]
		Assessment of polymerization	[83] *
		Interaction with SulA	[40] *
DLS	[54,78]	Assessment of polymerization	[80,82]
SV	[54]	Assessment of polymerization	[41]
		Interaction with SlmA	[93] *
FRET, intrinsic fluorescence	[63]	Assessment of polymerization	[30] *
		Interaction with FtsA, MinC	[94,95] *
Biosensor	[56]	Interaction with SlmA, MinC	[12,96] *
Fluorescence microscopy	[97]	Assessment of polymerization	[98]
		Interaction with ZapA	[75,99]
**Reconstruction systems**			
Nanodiscs	[100]	Interaction with ZipA	[37] *
Microbeads	[56]	Interaction with ZipA	[101] *
		Interaction of FtsA, SlmA with membrane	[102,103] *
SLBs	[55,57]	Interaction with MinCDE, FtsA, ZipA	[104,105,106,107,108] *
Biosensor, plasmonic sensor	[56]	Interaction with ZipA	[109,110] *
		Interaction of SlmA, MinDE with membrane	[103,111] *
Microdroplets, liposomes	[55,56,57,58,112]	Interaction with FtsA, ZipA	[113,114,115,116] *
		Interaction of MinCDE with membrane	[108]*
		Arrangement & distribution of FtsZ species	[116,117,118,119] *

## Data Availability

No new data were created or analyzed in this study. Data sharing is not applicable to this article.

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
