# Peer review of "FtsZ Interactions and Biomolecular Condensates as Potential Targets for New Antibiotics"

_antibiotics, 2021, doi:10.3390/antibiotics10030254_

Round 1

Reviewer 1 Report

The review by Zorrilla et al. is very well written, and presented in a logical fashion. The included figures aid in the discussion of key points in the manuscript.  This subject will be of interest to specialists and the broader scientific community.

I have one small comment.  For the benefit of a non-expert, it would be useful to describe a "condensate" in simpler terms (around line 96).

Reviewer 2 Report

The authors comprehensively elucidated the importance of the bacterial cell division cycle as target for antimicrobials considering FtsZ itself and its interactions with several others crucial proteins.
The whole paper provides a full insight of the division cycle, useful for several classes of researchers involved into overcoming antimicrobial resistance.

Nevertheless, a few adjustments could be performed:

  • In chapter 3, the authors often refer to compounds described in literature (compound 8j, compound 13 etc.) without reporting the chemical structure nor the molecular class to which they belong. The reader could be confused; therefore, I suggest to, at least, generate a very simple table reporting the main cited compounds.
  • In chapter 3 and 4 the authors list, respectively, the main techniques to identify molecules which interacts with FtsZ polymerization and the main interactions between FtsZ and others crucial cell division proteins.
    In chapter 3, it could be useful to generate subsections, each introducing a class of techniques (light scatterings, fluorescence-based etc.), giving a higher degree of accessibility to each information. In a similar manner, in Chapter 4, every FtsZ partner could be introduced by a subsection.

Besides these slight modifications, I recommend the paper for publication.

Reviewer 3 Report

This review is well written and composed. The information provided is comprehensive and also unique. I have only a couple of minor suggestions:

  1. Many abbreviations in the article should be expanded/spelled out in their first instance for the readers.
  2. Figure 1 provides a nice overview of TtsZ's interactions with other molecules. It will make referencing easier if the authors can include the sub-section (Fig.1a-1f) in the text when mentioning this information-dense figure.
  3. For Figure 3, if does not provide much more additional information than the text. I think it will be more helpful if this figure is converted into a table with examples (purpose and its reference) for each techniques/methods.
